# The Soil Seed Bank Role in Mountainous Heathland Ecosystems after Fire and Inorganic Nitrogen Fertilization

Josu G. Alday [1,2], Leonor Calvo [3], José Luis Fernández Rodríguez [3] and Luz Valbuena [3,*]

1 Joint Research Unit CTFC–AGROTECNIO–CERCA, 25198 Lleida, Spain
2 Department of Crop and Forest Sciences, University of Lleida, Av. Alcalde Rovira Roure 191, 25198 Lleida, Spain
3 Area of Ecology, Department of Biodiversity and Environmental Management, University of León, 24071 León, Spain
* Correspondence: luz.valbuena@unileon.es

**Abstract:** *Calluna* vulgaris-dominated heathlands are a priority habitat type in Annex I of the Habitats Directive (92/43/ECC, habitat code 4060). In the Iberian Peninsula, the landscape of the Cantabrian Mountain range has great heterogeneity due to human management during the last 10,000 years. Another factor that can affect these communities is the increase in human-induced atmospheric nitrogen (N) deposition. During the last century, there has been a dramatic increase in N deposition rates. For all these reasons, it is important to know the regeneration dynamics of the heathlands in the context of the disturbances that these communities currently face (i.e., N deposition, fire, and decrease in sheep grazing) in the Cantabrian Mountain range. In this study, we characterized the plant species composition and soil seed bank after prescribed burning in three heathlands on their southern distribution limit in Spain, to gain insights into regenerative capacity and conservation of these communities. The results obtained suggest that the post-burn soil seed bank could restore *Calluna*-dominated vegetation in these habitats, indicating that the restoration potential from the soil seed bank after wildfires of these habitats is high. Our results also suggest that, in the short term after burning, the main characteristic species such as *Calluna* and *Erica* are recovered, which is fundamental to maintain the heathland community structure.

**Keywords:** heathlands; fire; inorganic nitrogen; seed bank; restoration; *Calluna vulgaris* (L) Hull.; diversity; ecosystem management

## 1. Introduction

European heathlands that are dominated by *Calluna vulgaris* are human-shaped habitats that play a crucial role in the traditional landscape, with widely recognized strong cultural, natural, and ecological values [1]. Heathland landscapes, which were widespread at the end of the 18th century from the north of Europe to the southern part of Spain [2], have reduced their extension during the last decades by the abandonment of human management [3]. Consequently, *Calluna vulgaris*-dominated heathlands are a priority habitat type in Annex I of the Habitats Directive (92/43/ECC, habitat code 4060: alpine and boreal heathlands) for conservation and protection of wild fauna and flora inside the European Union.

In the Iberian Peninsula, the landscape of the Cantabrian Mountain range has great heterogeneity due to human management during the last 10,000 years [4]. Therefore, *Calluna vulgaris*-dominated heathlands have an important representation in the foothills of this Mountain range. The traditional management of these plant communities has involved seasonal grazing and prescribed burning [4], which are somewhat interconnected. Grazing has been basically by sheep guided with a shepherd and, to a lesser extent, grazing by cows and horses [5]. It is well known that these lands were used during the summer for grazing by transhumant flocks of sheep from the south of Spain and by flocks from neighboring

villages. In contrast, traditionally, prescribed burnings have been carried out in those areas where the heathlands were very dense, preventing the entry of sheep and cattle [5]. Because of this traditional management, the *Calluna* heathlands have developed great floristic richness by adapting to recurrent disturbances, and therefore, are a well-known biodiversity hotspot [6]. For instance, in the heathlands of the Cantabrian Mountain range we can find more than 80 plant species, with *Calluna vulgaris* and *Erica tetralix* L. being the dominant species [7]. However, the abandonment of rural areas in recent decades has threatened the Spanish *Calluna* heathlands due to three forces: Firstly, common to other European regions, there has been an intense decrease in the number of cattle grazing in mountain areas [8], for instance, in 1832, there were 551,032 transhumant sheep in the Province of León, while in 2000, there were only 10,166 sheep [9]. Secondly, the type of livestock grazing has changed; nowadays, there has been an increase in the number of cows and horses and a reduction of sheep [9]. Thirdly, fire suppression has become a standard response to prevent prescribed burnings in many fire-adapted ecosystems [10]. Recently, there has been new interest in the management of *Calluna* heathlands in Spain, particularly, with respect to maintaining their conservation value and ecosystem services that they provide.

Fire and herbivores can both be considered to be evolutionary forces that have marked the particular traits of many plants and the evolution of biomes [11]. The plant species in *Calluna* heathlands have evolved with these forces, and have adapted to livestock grazing and a regime of human-induced fire. It is important to mention that these evolutionary forces not only determine the plant species living in heathlands but also the most common plant regeneration strategies after disturbance. For example, *Calluna vulgaris* is a well adapted germinated species after fire [8], although, in the North German lowlands, some regrowth capacity in young plants has also been described [12,13]. Consequently, between prescribed fire disturbances most heathland plant species maintain their seeds in the soil, forming the soil seed banks that allow plant community recovery after fire [14]. Soil seed banks have a dual function in terrestrial ecosystems: They are memories of populations and communities past, and they also provide the potential for future persistence and survival [15]. Currently, it remains unclear whether the *Calluna* heathlands soil seed bank will respond positively or negatively to land abandonment and land use changes in the Spanish Cantabrian Mountain range, especially from a cultural and ecosystem conservation perspective.

Another factor that can affect these communities is an increase in human-induced atmospheric reactive nitrogen (N). During the last century, there has been a dramatic increase in N deposition rates [16]. Ecosystems that have adapted to low levels of nutrient availability, such as *Calluna* heathlands, are particularly sensitive to airborne N deposition, especially in north-western and central Europe [17]. However, in the Cantabrian Mountain range heathlands, Calvo et al. [5] found that there were no significant changes driven by N deposition in the composition and richness of plant species present. The different plant functional groups showed contrasting responses to the cumulative addition of N: Annual/perennial forbs and annual graminoids increased with quite low values and perennial graminoids were rather abundant in young heathlands but only slightly augmented in mature ones [18]. Thus, for conservation guidelines, it would be necessary to assess changes in the main heath species (*Calluna* and *Erica* spp.), testing whether the regeneration pattern of these two plants has been conditioned by N fertilization.

For all these reasons, it is important to know the regeneration dynamics of heathlands in the context of the disturbances that these communities currently face (i.e., fire exclusion and decrease in sheep grazing) in the Cantabrian Mountain range. This knowledge would help with decision making to design adaptive management for protecting heathlands under the Habitats Directive of the European Union. In this paper, we characterized the plant species composition and soil seed bank in three heathlands on their southern limit in Spain to gain insights into regenerative capacity and conservation of these communities. Here, our main aims were to: (1) Characterize the plant species composition and diversity of these

sites before any disturbance. (2) Describe the soil seed bank composition and diversity considering the main two soil layers before and after a disturbance and related to plant species composition. (3) Describe the *Calluna* and *Erica* seedlings establishment, in the short and medium term, after prescribed fires and in response to N fertilization. We checked the soil seed bank differences between the three heathlands in the number of seeds stored, species richness, and evenness. The assessment of plant and seed bank attributes is the first step towards the knowledge of the heathlands resistance to perturbations (e.g., wildfires) and to determine whether natural regeneration will be sufficient for maintaining these ecosystems.

## 2. Materials and Methods

### 2.1. Study Area

The study area is located on the southern slope of the Cantabrian Mountain range (NW Spain). Here, three representative and homogeneous *Calluna* heathland sites were selected: Riopinos I (RIoP1) (1660 m a.s.l., 43°02′ N, 5°24′ W) is a discontinuous northern-exposed heathland on a steep slope; Riopinos II (RioP2) (1560 m a.s.l., 43°02′ N, 5°26′ W) is a wind-exposed heathland in a north facing area with a low slope; San Isidro (SanIsi) (1620 m a.s.l., 43°03′ N, 5°21′ W) is a flat and continuous heathland facing north and exposed to winds [19]. The soils for the three sites are Umbrisol (European Commission, 2005), and are characterized by sandy texture, high acidity (pH = 3.9 ± 0.14, deionized water), and low fertility. Occasionally, these soils are subjected to a podzolization process [20]. Soil depth ranges from 30 to 50 cm (on sandstone and lutite).

The average annual temperature in these areas is 5.5 °C, while the average annual rainfall is 1645 mm, mainly in the form of snow, distributed from October to May. These areas also have two dry summer months (July and August). The climatic characteristics in these areas restrict the vegetation growth period, being reduced to 4 or 5 months (from June to October). The dominant vegetation is *Calluna vulgaris*, (80% cover, with 20 cm height in young stands and 50 cm height in mature stands), with *Erica tetralix* L. and *Vaccinium myrtillus* L. as the main accompanying species. These heathland formations represent the southernmost examples of habitat 4060 in Western Europe [21].

As other heathlands in the world [22], these areas are subjected to N depositions. For instance, the bulk inorganic N deposition from 2011 to 2014 was 4.6 kg N ha$^{-1}$ yr$^{-1}$, and the total N deposition ranged between 7.5 and 15 kg N ha$^{-1}$ yr$^{-1}$ (in 5 years averages), according to the EMEP and CHIMERE models for Spain [23]. Therefore, total N deposition in the study area is either lower than or within the lowest critical load value estimated to threaten the persistence of European dry *Calluna* heathlands (i.e., 10–20 kg N ha$^{-1}$ year$^{-1}$ [19]).

### 2.2. Experimental Design

In each of the 3 selected sites, in 2004, we randomly selected four permanent blocks (20 × 10 m). Each of the blocks were subjected to different treatments: (1) control plot, (2) fertilized plot, (3) burned plot, and (4) burned plus fertilized plot. The experimental burns were done between June and July 2005, for that, the Forestry Service measured environmental data such as air temperature, relative moisture, and wind speed in 2005 (Table 1). Average fuel loading was also estimated in each study site for the two most representative species (*C. vulgaris* and *E. tetralix*) previous burnings. The aim was burning all sites under similar conditions.

Fertilization was carried out with ordinary granules of ammonium nitrate, spread manually every month from July to October 2005 and from June to October 2006, 2007. The total concentration of fertilizer (56 kg N ha$^{-1}$ year$^{-1}$) was equivalent to four times the maximum deposition levels in this area and application was equally divided over these months. The total N deposition ranged between 7.5 and 15 kg N ha$^{-1}$ year$^{-1}$, according to the EMEP and CHIMERE models for Spain [23].

**Table 1.** Climatic data and average fuel loading at the time of ignition in each of the three selected heathlands in the Cantabrian Mountain Range in 2005 (Spain).

|  | Riopinos I | Riopinos II | San Isidro |
|---|---|---|---|
| Relative moisture (%) | 40 | 56 | 62 |
| Wind speed (m s$^{-1}$) | 2.4 | 1.6 | - |
| Air temperature (°C) | 20 | 17 | 16 |
| Average fuel loading (kg m$^{-2}$) |  |  |  |
| *Calluna vulgaris* | 2.02 | 2.10 | 1.95 |
| *Erica tetralix* | 0.00 | 0.17 | 0.10 |

*2.3. Vegetation Analysis*

Prior to any treatment in June 2004, we randomly selected ten 1 × 1 m permanent sampling plots in each block and site. In each plot, the cover of all vascular plant species was visually estimated. At the same time, after the burning treatment application (burned and burned plus fertilized plots) we developed an emergence test by counting the number of *C. vulgaris* and *E. tetralix* seedlings during the vegetative period (June–October) from 2005 to 2007. For that, we used the permanent sampling plots of each block, and marked and counted the seedlings of the two species that germinated and survived. Nevertheless, in 2008, 2009, and 2012, the emergence sampling was reduced to June and September. It is important to highlight that during the first two years after burning (2005 and 2006) there were no seedlings of the two target species (*C. vulgaris* and *E. tetralix*), which started to regenerate again in 2007.

*2.4. Soil Seed Bank Analysis*

For the soil seed bank analysis, we followed the definition by Leck et al. [24] which described the soil seed banks as all viable seeds in or on the ground, including litter. Therefore, in each block of each site, 160 cores of 98.17 cm$^3$ were randomly collected using a hollow steel cylinder that was 5 cm high by 5 cm in diameter. It is well known that 90% of the total population of buried heather seeds lies within the top 5 mm of podzolic soils [25]. Afterwards, in the laboratory, each sample was subdivided into two strata: the upper stratum from 0 to 2.5 cm and the lower stratum from 2.5 to 5 cm.

The soil samples were taken before (2004) and after (2008) experimental treatments. First, in autumn 2004, the before-soil seed bank monitoring was done. We selected autumn because it was assumed that, at this period, the contribution of seeds to the soil had been made. In September 2008, the collection of samples was repeated to analyze the effect of burning and fertilization on the soil seed bank (after-soil seed bank). We followed the same methodology for the pre- and post-burn soil seed bank determination.

The soil seed bank was estimated using an indirect method, i.e., the seedling emergence technique, for which the 160 soil cores were mixed according to strata (upper and lower) and separated into 20 samples per stratum and block. Each of these samples were spread forming a thin layer inside a tray of 30 × 40 cm and 5 cm thick on two layers of inert substrate: one lower layer of gravel and the other upper layer of sand. The samples distributed in trays were placed inside a chamber under controlled conditions following the methodology of [26]. The soil seed bank emergence experiments lasted one year each (2004 and 2008). Emerging seedlings were identified, counted, and removed; when the identification was uncertain, the seedling was transplanted into a pot and grown until an accurate identification could be made.

*2.5. Data Analysis*

Statistical analyses were performed in the R software environment (R Core Team, 2021), using the "vegan" package for multivariate and diversity analyses [27], and linear mixed models (LME) were used for comparison of richness, evenness, bare soil, and the seed bank density ('nlme' package Pinheiro et al. 2012). The 'emmeans' package (v1.4.1; [28]) was

used for post hoc pairwise comparisons between sites. The seedling models were done using general additive mixed models (GAMM, 'mgcv' package [29]).

Vegetation community data were analyzed using both multivariate and univariate analysis methods to characterize the initial differences between heathlands. First, to describe the differences in plant species composition between sites, we used permutational multivariate analysis of variance performed with the function "adonis" [27]. Here, the block structure of the experiment was included to limit randomizations. In addition, differences in species composition were tested using Bray–Curtis distance [30]. Plant cover values were initially log-transformed (log(cover+1)) to reduce the influence of abundant species [30]. Afterwards, we performed a NMDS using Bray–Curtis dissimilarities, plotting over the ordination the site standard deviational ellipses ("ordiellipse" function). At the same time, species detected were classified into two life forms: herbaceous species and woody species. The univariate analysis method was used to test the differences in richness, evenness, and bare soil between sites. We used linear mixed models (LME) considering the site (RioP1, RioP2, and SanIsi) as the fixed factor and plots nested with blocks ($n$ = 4 per site) as the random factor to account for spatial autocorrelation. When LMEs were significant, the Tukey method for pairwise comparisons were used to visualize differences between sites.

Soil seed bank data were also analyzed using both multivariate and univariate analysis methods to characterize the initial differences between heathlands. The pretreatment soil seedbank data (2004) were subjected to three PMAVs with Bray–Curtis distance and 999 permutations; overall and split in two soil layers, i.e., upper and lower. Quantitative shifts in the soil seed bank were tested using the species-abundance matrix Hellinger transformed [27]. Thereafter, NMDS with Bray–Curtis distance was used to observe in ordination space the seed bank compositional differences between sites. To help the interpretation, standard deviational ellipses of each site were then used to illustrate the positions of sites on the biplots. The univariate analysis method was used to test the differences in soil seed bank richness, evenness, and seed densities between sites. In these models, we followed the same structure described above. Unfortunately, the post-burn soil seed bank experiment (2008) and data gathering were not strong enough to be able to use in multivariate methods nor in linear models, thus, we will describe the main results found for burn and burn-fertilized blocks.

The *Calluna* and *Erica* seedlings development in burned plots was analyzed at two levels, i.e., intra-annually during the 2007 growing season and inter-annually from 2007 to 2012, using generalized additive mixed models (GAMMs). We chose GAMMs since they make no a priori assumption about the functional relationship between variables [29]. Here, the dependent variable was fitted with respect to elapsed time (months), with elapsed time nested within plots and sites to account for temporal correlation. Five knots were used for the regression splines in order avoid overfitting the data, but still allowing for unimodal or slightly more complex model fits [29]. The lack of germinated seedlings of both species during 2005 and 2006 made their use impossible. A comment on it was included.

## 3. Results

A total of 27 plant species were described; 23 plant species in SanIsi, 7 plantspecies in RioP1, and 15 plant species in RioP2. Splitting these values by woody and herbs, 10 woody species were found with dominating cover values (>70%), while there were 8 herbaceous species with cover values around 10%. The most abundant woody species were *C. vulgaris* (mean relative coverage 67%), *E. tetralix* (20%), and *Vaccinium myrtillus* (7%), while the most abundant herbs were *Nardus stricta* (3.6%), *Juncus capitatus* (1.8%), and *Deschampsia flexuosa* (1.4%) (Table 2).

**Table 2.** Initial vegetation survey results show the mean plant cover (%± standard deviation) of woody species and more abundant herbs in the three selected heathlands in the Cantabrian Mountain Range (Spain).

| Sites | San Isidro | Riopinos I | Riopinos II |
|---|---|---|---|
| UTM: 30TUN | 3082 47694 | 3035 47687 | 3007 47685 |
| Altitude (m a.s.l.) | 1620 | 1660 | 1560 |
| Woody species | | | |
| *Calluna vulgaris* L. (Hull) | 65.7 ± 4.1 | 67.9 ± 5.0 | 67 ± 5.5 |
| *Genista tridentata* L. | 0.2 ± 0.31 | - | - |
| *Cytisus oromediterraneus* Rivas Mart. & al. | 0.3 ± 0.4 | - | - |
| *Erica australis* L. | - | 5.5 ± 8.3 | 0.1 ± 0.3 |
| *Erica tetralix* L. | 24.1 ± 16.2 | 0.2 ± 0.5 | 36.0 ± 18.2 |
| *Ulex minor* Roth | - | - | 1.5 ± 3.0 |
| *Vacciniun myrtillus* L. | 14.6 ± 5.3 | 2.7 ± 3.2 | 2.2 ± 1.1 |
| *Vacciniun uliginosum* L. | 4.4 ± 6.1 | - | - |
| *Polygala microphylla* L. | 0.2 ± 0.4 | - | 0.02 ± 0.05 |
| Herbs | | | |
| *Carex muricata* L. | 1.1 ± 1.3 | - | 0.2 ±0.3 |
| *Deschampsia flexuosa* (L.) Trin. | 2.6 ± 0.8 | 0.6 ± 1.2 | 0.9 ± 0.9 |
| *Festuca rubra* L. | 0.7 ± 1.1 | 0.5 ± 0.6 | - |
| *Hieracium pilosella* L. | 0.1 ± 0.2 | - | - |
| *Jasione montana* L. | 0.5 ± 0.9 | - | - |
| *Juncus squarrosus* L. | 2.9 ± 3.2 | 0.3 ± 0.5 | 2.2 ± 2.9 |
| *Nardus stricta* L. | 5.1 ± 0.9 | - | 5.9 ± 4.4 |
| *Potentilla erecta* (L.) Raeusch. | 1.3 ± 1.2 | - | 1.3 ± 2 |
| Life forms | | | |
| Woody | 109.5 ± 13.9 | 75.1 ± 1.9 | 106.8 ± 12.3 |
| Annual herbs | 0.5 ± 0.9 | - | - |
| Perennial herbs | 17.1 ± 4.7 | 1.3 ± 2.3 | 10.6 ± 8.5 |
| Mean % of Bare soil | 51.2 ± 5.4 | 60.2 ± 3.4 | 38.9 ± 5.6 |

*3.1. Initial Vegetation Differences between Sites*

There was a significant site effect on plant species composition, which explained around 45% of the compositional differences (PMAV, F = 47.01, $p < 0.001$, $R^2$ = 0.45). NMDS ordination (stress 0.15, Figure 1) and standard deviation ellipses clearly showed that the RioP2 and SanIsi sites were grouped at the positive end of Axis 1 with no significant differences between them (Figure 1a). In contrast, the RioP1 site was located at the negative end of Axis 1, showing clear compositional differences as compared with the other two sites. Focusing on the most important species (Figure 1b), the SanIsi site was characterized by the presence of *Agrostis capillaris*, *Jasione montana*, *Vaccinium myrtillus*, and *Viola riviniana* Reichenb, which had more herbaceous species than the other two sites (Table 2). *Erica tetralix*, *Nardus stricta,* and *Ulex minor* were abundant species at the RioP2 site, while the RioP1 site was more related to *Erica australis* and *Conopodium majus* (Figure 1b). The two most abundant species, *C. vulgaris* and *D. flexuosa*, were in the center of the ordination.

Regarding diversity variables, there was a site effect over plant species richness ($F_{[2,78]}$ = 114.55, $p < 0.001$). The SanIsi site showed greater species richness than the RioP1 and RioP2 sites (six, four, and two mean species, respectively), mainly produced by the high number of herbaceous species (eight species) as compared with the RioP sites (three and five species for the RioP1 and RioP2 sites, respectively, Table 2). There were evenness differences between sites ($F_{[2,78]}$ = 58.68, $p < 0.001$). However, the evenness values were similar for the RioP2 and SanIsi sites (0.65 and 0.68, respectively, paired test $p$ = 0.827). In contrast, there was a strong dominance of *C. vulgaris* in the case of the RioP1 site, showing significantly lower evenness values (0.23). Finally, regarding bare soil, there were significant differences among sites ($F_{[2,78]}$ = 44.32, $p < 0.001$), with the highest bare soil value (60.25)

at RioP1 ), followed by the SanIsi site (51.25), and the lowest bare soil value at the RioP2 site (38.88).

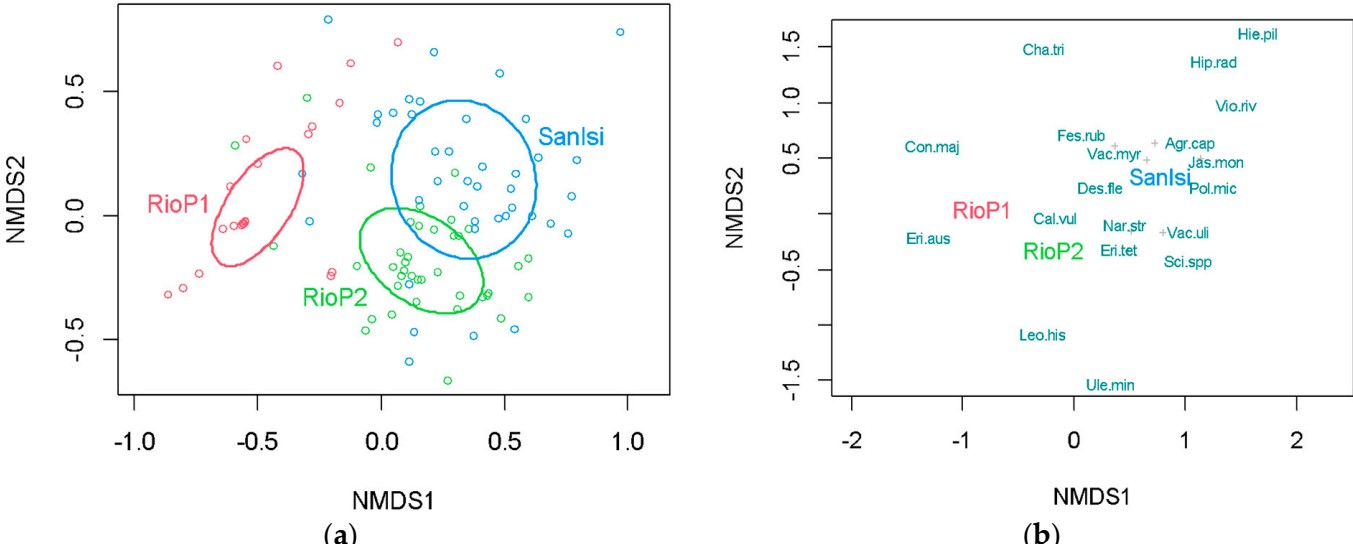

**(a)**　　　　　　　　　　　　　　　　　　　**(b)**

**Figure 1.** NMDS ordination for the first two axes of vegetation compositional differences between the three heathlands before any treatment application in the Cantabrian Mountain range, illustrating: (**a**) Ordination biplot with deviational ellipses for the three sites; (**b**) species biplot where only species with greater cover are shown. Less important species are represented as grey crosses. Species codes are: Agr.cap, *Agrostis capillaris*; Cal.vul, *Calluna vulgaris*; Cha.tri, *Genista tridentata*; Con.maj, *Conopocium majus*; Des.fle, *Deschampsia flesuosa*; Eri.aus, *Erica australis*; Eri.tet, *Erica tetralix*; Fes.rub, *Festuca rubra*; Hie.pil, *Hieracium pilosella*; Hip.rad, *Hypochoeris radicata*; Jas.mon, *Jasione montana*; Leo.his, *Leontodon hispidus*; Nar.str, *Nardus stricta*; Pol.mic, *Poligala microphylla*; Sci.spp, *Scilla* spp.; Ule.min, *Ulex minor*; Vac.myr, *Vaccinium myrtillus*; Vac.uli, *Vaccinium uliginosum*; Vio.riv, *Viola riviniana*.

*3.2. Soil Seed Bank Differences among Sites*

The analysis of the overall soil seed bank (Table 3) showed that there were significant differences among sites, which explained 52% of the compositional variance (PMAV, F = 127.57, $p$ = 0.001; $r^2$ = 0.52). The NMDS ordination (stress 0.14, Figure 2a) clearly showed that the three sites were displaced across Axis 1, the RioP1 site was located at the negative end and was significantly different from the other two sites, while the SanIsi site was in a central position and the RioP2 site was at the positive end but not different from the SanIsi site (Figure 2a). When the data were split into upper and lower layers, similar results were found. The upper layer showed greater differences among the sites (PMAV, F = 101.18, $p$ = 0.001; $R^2$ = 0.46), while the lower layer showed lower differences but still significant (PMAV, F = 43.83, $p$ = 0.001; $R^2$ = 0.29). In both cases, the distribution of sites was identical across Axis 1 and the main differences were between the RioP1 and RioP2 sites, since the SanIsi site maintained a central position in both the upper and lower soil layers.

There were significant differences in seed densities among sites considering each of the two layers (upper: $F_{[2,158]}$ = 41.10, $p$ < 0.001 and lower: $F_{[2,158]}$ = 24.34, $p$ < 0.001). In both cases, the RioP1 site showed significantly lower seed densities in the two soil layers than the RioP2 and SanIsi sites (upper 109 vs. 212 seeds/m$^2$ and lower 31 vs. 88 seeds/m$^2$). The differences in seed bank densities were related to the species that can be found in each site; for example, while SanIsi and RioP2 were characterized by species such as *Juncus squarrosus, Daboecia cantabrica, Erica tetralix,* and *Jasione montana* (Figure 2b), there were few species related to the RioP1 site, being dominated by *C. vulgaris* and *Agrostis capillaris*. This also had an influence on the soil seed bank richness with significantly lower values in the RioP1 site than in the other two sites for the two layers considered (upper: $F_{[2,158]}$ = 109.65, $p$ < 0.001 and lower: $F_{[2,158]}$ = 44.35, $p$ < 0.001). The mean species richness for the upper

and lower soil layers of RioP1 was 1, while for RioP2 and SanIsi the richness values were 3 in all cases. Similar results were obtained for soil seed bank evenness ($p < 0.001$), with the RioP1 site showing significantly lower values than the other two sites, which had a clear dominance of *C. vulgaris* seeds in this site.

**Table 3.** Overall description of the soil seed bank after burning in 2005 in the three selected heathlands in the Cantabrian Mountain range (Spain).

| Sites | San Isidro | | Riopinos I | | Riopinos II | |
|---|---|---|---|---|---|---|
| UTM: 30TUN | 3082 47694 | | 3035 47687 | | 3007 47685 | |
| Altitude (m. asl.) | 1620 | | 1660 | | 1560 | |
| Species/seed/m$^2$ | Upper Stratum | Lower Stratum | Upper Stratum | Lower Stratum | Upper Stratum | Lower Stratum |
| Woody species | | | | | | |
| *Calluna vulgaris* | 18,100.00 | 5353.125 | 9725.00 | 2750.0 | 13,821.88 | 4253.125 |
| *Erica australis* | 46.88 | 6.25 | 3.13 | - | 3.13 | - |
| *Erica tetralix* | 3.13 | 12.5 | 9.38 | - | 31.25 | 9.375 |
| *Daboecia cantabrica* (Huds.) K.Koch | 806.25 | 159.375 | 3.13 | 3.125 | 1,062.50 | 787.5 |
| *Vaccinium myrtillus* | 3.13 | - | 6.25 | 6.25 | 9.38 | 6.25 |
| *Genista* sp. | 3.13 | - | - | - | - | - |
| Herbs | | | | | | |
| *Carex* sp. | 31.25 | 40.625 | 6.25 | - | 3.13 | 18.75 |
| *Juncus squarrosus* | 690.63 | 2206.25 | 40.63 | 21.875 | 1462.50 | 1253.125 |
| *Agrostis capillaris* L. | 575.00 | 343.75 | 6.25 | 9.375 | 1675.00 | 1300 |
| *Rumex acetosella* L. | 3.13 | 31.25 | 28.13 | 12.5 | 12.50 | 6.25 |
| *Galimu saxatile* L. | 6.25 | 6.25 | - | - | 28.13 | 100 |
| *Geranium* sp L. | 0.00 | 3.125 | 3.13 | 3.125 | - | - |
| *Potentilla erecta* | 0.00 | 3.125 | - | - | - | - |
| *Potentilla erecta* | 0.00 | 3.125 | - | - | - | - |
| *Leontodon pyrenaicus* Gouan | 18.75 | 15.625 | - | - | - | - |
| *Sonchus* sp. L. | 0.00 | 3.125 | - | - | - | - |
| *Thymelaea* sp. Mill | 3.13 | - | - | - | - | - |
| *Jasione montana* | - | - | - | - | - | 3.125 |

The post-treatment soil seed banks for the RioP1 and RioP2 sites were similar, due to the exclusive presence of *C. vulgaris* and *A. capillaris* in the burn and burn + fertilized blocks, with seed densities of 31 and 3 seeds/m$^2$, respectively, for both sites. In contrast, the SanIsi site showed greater seed bank richness after the treatments, with four species in both blocks. Here, the soil seed bank was still dominated by *C. vulgaris* (281 seeds/m$^2$) followed by *A. capillaris* (16 seeds/m$^2$) and there were three new species present, i.e., *Carex* spp. (13 seeds/m$^2$), *R. acetosella* (6 seeds/m$^2$) and *J. squarrosus* (3 seeds/m$^2$).

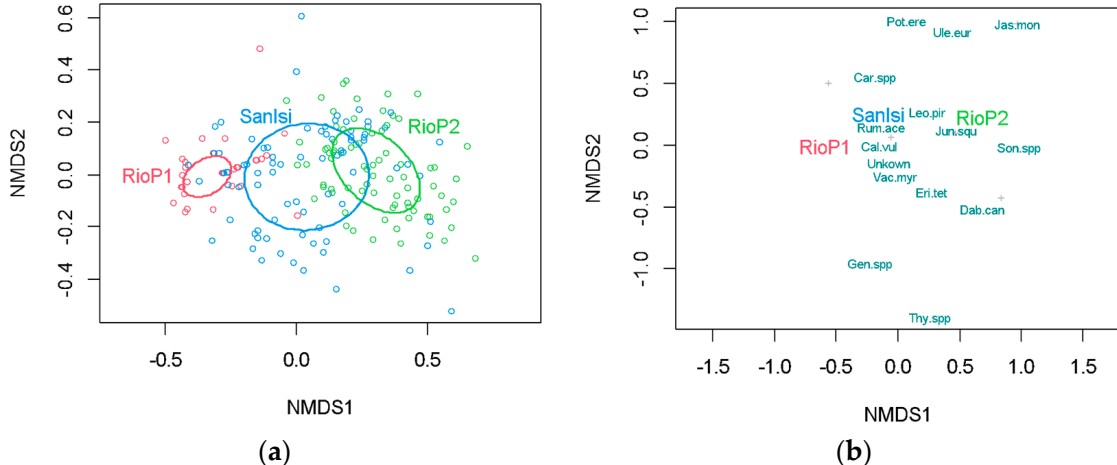

**Figure 2.** NMDS ordination for the first two axes of pretreatment soil seed bank compositional differences among the three heathlands before any treatment application in the Cantabrian Mountain range, illustrating: (**a**) Ordination biplot with deviational ellipses for the three sites; (**b**) species biplot where only species with high seed densities are shown. Less important species are represented as grey crosses. Species codes are: Cal.vul, *Calluna vulgaris*; Car.spp, *Carex* spp.; Dab.can, *Daboecia cantabrica*; Eri.tet, *Erica tetralix*; Gen.spp, *Genista* spp.; Jas.mon, *Jasione montana*; Jun.squ, *Juncus squarrosus*; Leo.pir, *Leontodon pyrenaicus*; Pot.ere, *Potentilla erecta*; Rum.ace, *Rumex acetosella*; Son.spp, *Sonchus* spp.; Thy.spp, *Thymelaea* spp.; Ule.eur, *Ulex europaeus*; Vac.myr, *Vaccinium myrtillus*.

### 3.3. Vegetation and Soil Seed Bank Comparisons

By comparing the vegetation and the soil seed bank composition, it can be observed that there were similarities in composition and abundance of each species. For example, at the SanIsi site, the dominant plant species were *Calluna vulgaris*, *Erica tetralix*, *Agrostis capillaris*, *Jasione montana*, and *Vaccinium myrtillus*, while the soil seed bank was dominated by *Calluna vulgaris*, *Erica tetralix*, *Vaccinium myrtillus*, *Rumex*, and *Carex* spp. A similar situation was observed at the RioP2 site with *Calluna vulgaris*, *Erica tetralix*, *Nardus stricta*, and *Ulex minor* as the main vegetation species and with a soil seed bank dominated by *C. vulgaris*, *E. tetralix*, and *Agrostis capillaris*. The only difference found was at the RioP1 site where the soil seed bank was dominated by *C. vulgaris* and *Erica australis*. In any case, after the burned plus fertilized treatments at the three study sites, the soil seed bank maintained an important amount of seed densities from *C. vulgaris*, *A. capillaris*, *E. tetralix* and *E. australis*.

### 3.4. Seedlings Establishment after Controlled Fire

The number of *C. vulgaris* and *E. tetralix* seedlings germinated in the burned plots during the vegetative period of 2005 and 2006 were nearly zero individuals at the three sites. Only during the vegetative period of 2007, there was germination of these two woody species. The overall GAMMs for *Calluna* seedlings for 2007 showed that there was a significant time × treatment interaction (F = 35.55, $p < 0.001$). The number of *Calluna* germinated seedlings increased significantly through the time in burn and burned + fertilized treatments (Figure 3a); however, in the burned + fertilized treatment there was a lineal increase, arriving at 80 seedlings per m$^2$ in October of 2007. In contrast, the burned treatment showed a slower increase during the first months, June–July, and afterwards, showed a step increase, reaching a maximum of 110 seedlings per m$^2$ in October. Although the germination trends were quite similar in the three sites, there were significant differences among the sites in the number of *Calluna* seedling that germinated during the vegetative period, with lower mean germinated *Calluna* seedlings at the RioP2 site as compared with the other sites (29 at the RioP2 site vs. 51 at the RioP1 site and 61 at the SanIsi site).

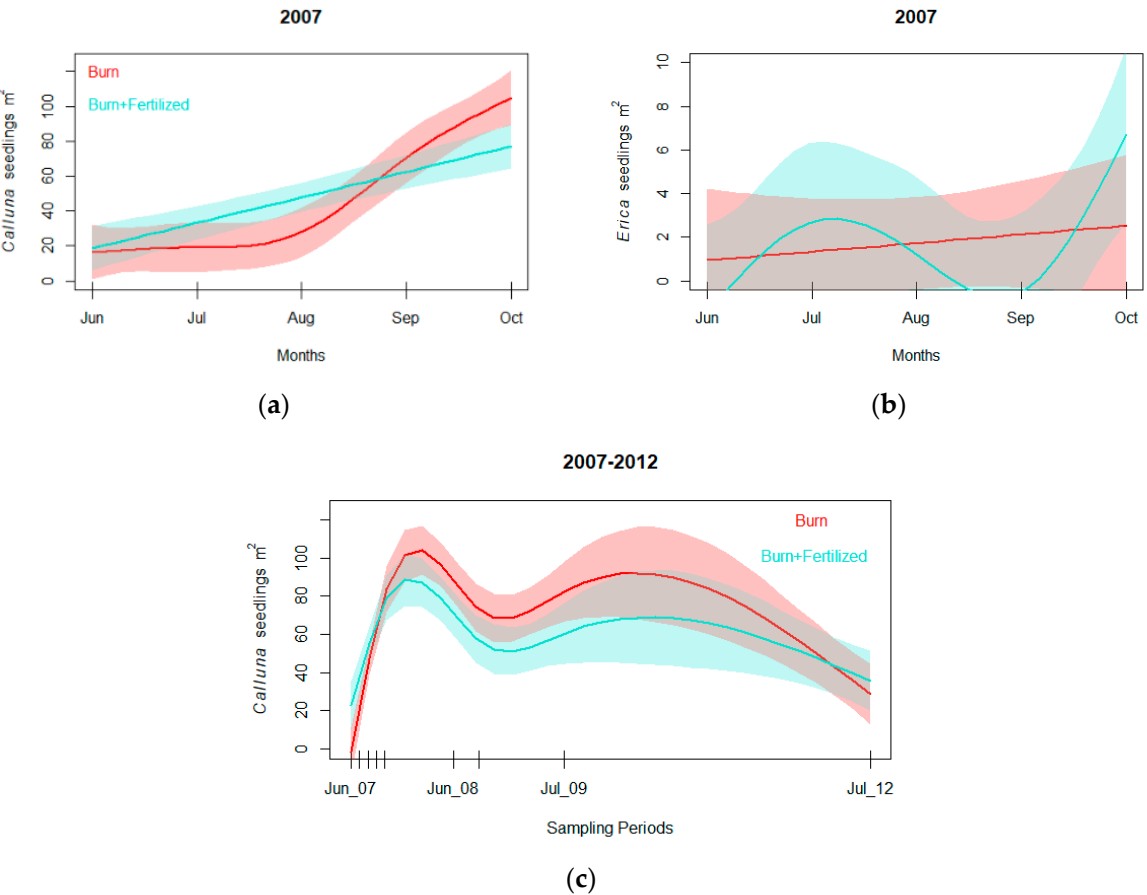

**Figure 3.** Germinated seedlings number for *Calluna* and *Erica* after burning of heathlands in the Cantabrian Mountain range during the growing season of 2007 (June–October) and for *Calluna* during all the studied period (2007–2012). The fitted line represents the best GAMM and the 95% confident intervals: (**a**) *Calluna vulgaris* seedlings in 2007; (**b**) *Erica tetralix* seedlings in 2007; (**c**) *Calluna vulgaris* seedlings across all the studied period. Red, burn treatments; turquoise, burned + fertilized treatment.

The overall GAMMs for *Erica* seedlings during the vegetative period of 2007 also showed that there was a significant time × treatment interaction (F = 3.51, $p$ = 0.015). The number of *Erica* seedlings increased significantly across the time in burn treatment (lineal, Figure 3b), reaching a maximum of 2 seedlings per m$^2$ in October 2007. However, in the burned + fertilized treatment, the number of germinated *Erica* seedlings followed a pulse trend. Interestingly, there were two increases between late June and early August and between late September and October. In contrast, during August, the germinated seedlings decreased (Figure 3b). The maximum number of *Erica* seedlings in the burned + fertilized treatment was in October 2007 with 6 seedlings per m$^2$. Regarding the site effect, the *Erica* germination trends were lineal for the RioP1 and RioP2 sites in both burn treatments, with low germinates as compared with those at the SanIsi site (1 and 2 seedlings per m$^2$ for RioP1 and RioP2 vs. 6 seedlings per m$^2$ for SanIsi). The main burned + fertilized pulse trend was mainly produced by the SanIsi site, because it had the greater number of germinated *Erica* seedlings.

When the inter-annual trends were evaluated (2007–2012) the GAMM results showed that, for *Calluna* seedlings, there was only a significant time effect (F = 11.03, $p$ < 0.001) with similar response shapes in both treatments. There was an important increase in germination during 2007, reaching values over 100 seedlings per m$^2$, followed by a decrease until 2008, when the values reached a kind of plateau around 60 seedlings per m$^2$ with a final decrease in both treatments to values below 40 seedlings per m$^2$. The GAMM results for *Erica*

showed a non-significant time effect with a constant germination value for both treatments (F = 1.046, *p* = 0.307) around 2 seedlings per m$^2$.

## 4. Discussion

In this study, we have described the vegetation composition and the soil seed bank in *Calluna*-dominated heathlands in the Cantabrian Mountain range in Spain under experimental conditions designed to describe the effects of burning and nitrogen fertilization in the vegetation recovery. Overall, our results showed that, although the plant community composition, diversity, and soil seed bank were slightly different among the study sites, mainly because the SanIsi site had more herbaceous species after the burning treatments, the overall recovery process was similar in the three sites. In any case, just after fire, the soil seed bank showed a lack of *Calluna* and *Erica* seedlings germination. It seems that some time after a fire event is needed to start the recovery of the vegetation from the seed bank, and therefore, these months after a fire are the most susceptible for a heathland community [8]. However, in 2007, there was an important increment in *Calluna* seedling establishment with no clear effect of fertilization. *Erica* also showed some germination, although with constant values of around 2 seedlings per m$^2$. Interestingly, these results suggest that, even with existing differences in vegetation and soil seed bank composition, the recovery of the two main species in these ecosystems is guaranteed in both burned and burned + fertilized treatments, as seedling establishments are maintained until five years after the burns were done (2012).

In general, *Calluna vulgaris* showed very good regeneration in the three study sites, however, the main differences were associated with the coverage of *Erica tetralix* at the RioP1 site and greater richness of herbaceous plants at the SanIsi site. The similarity of vegetation at these sites was expected as local species pools and main ecosystem species are similar for them [8]; however, the high abundance of herbaceous species at the SanIsi site, associated with the plant diversity differences, could be an indicator of a community changing to an alternative state dominated by herbaceous species if some disturbances cause the system to cross the tipping point [31]. Nevertheless, the community composition at the three sites has typical species from heathlands with important conservation interest, such as *Calluna vulgaris*.

The vegetation differences described among the three sites are maintained in the pre-burning soil seed bank. Here, a greater similarity between the RioPII and SanIsi seed banks was found which, in turn, differed from the RioPI, site that had fewer species present in the soil seed bank. Nevertheless, of particular note is the presence of *Calluna* and *Erica* in the post-burning seed bank of the three sites, being key species for the maintenance of these communities [9]. The most abundant species in the pre-burning soil seed bank was *C. vulgaris* which had mean seed densities of 13,880 seeds/m$^2$ in upper and 4119 seeds/m$^2$ in the lower horizon. These values are in the upper range of those reported for upland soils, i.e., 1000–39,000 seeds/m$^2$, in Great Britain [32–34], which guarantees an optimal recovering of *Calluna* cover in a few years after burning.

One of the most important results of the soil seed bank before burning treatments was the identification of the low species richness of the propagule bank found on these heathlands, with only 15 species in overall. These results are similar to those reported by [34] in heathlands of UK, where a low number of plant species was found in the soil seed bank. It is possible that this low richness could be related to the necessity of some pre-cues to break dormancy of some seeds, for example, smoke or fire [35]. However, the post-burn soil seed bank experiment was not successful in describing most of the seeds present in the soil after burning. This could be caused by: (i) because the seed bank was collected in 2008, 4 years after the fire treatment, and during this time the seed bank had been depleted as most of the seeds were germinated and no new seeds arrived [34] or (ii) because the fire intensity was too high and prevented the germination of species that were not adapted to it [35]. At this point, further information is needed on seed banks before and just after fire on these heathlands to identify possible revegetation or management problems.

At the three sites studied, the dominant vegetation species are also dominant in the seed banks of the soil. This coincides with the theory of [15] where they consider that seed banks act as refuges and not as spillover for many of the common heathland species. This implies that seed banking is an important life history strategy in the system, and that seed banks cause, rather than follow, the floristic trends seen in post-fire vegetation. Thus, management of this area should focus on preserving this soil seed bank component to guarantee the natural recovery of the heathlands after any disturbance [36].

The seedling establishment after controlled fire was positive for revegetation purposes in both burned and burned + fertilized treatments, with a high number of seedlings germinating in all zones after two years of burnings. However, the recruitment of *C. vulgaris* seedlings was delayed almost 2 years after fire. Interestingly, the *C. vulgaris* seedling recruitment after burned and burned + fertilized treatments was different; in burned + fertilized treatment there was a linear increase during the year, reaching 65 seedlings per $m^2$, while the burned treatment showed a slower *C. vulgaris* seedling increase during the first months, i.e., June–July, and then showed a staggered increase from August to October, reaching almost 100 *C. vulgaris* seedlings per $m^2$. Regarding *Erica,* there was a steady increasing trend with some peaks in fertilized plots but not significant from a restoration point of view. At least the number of *Erica* seedlings found during 2007 was stable around 2 per $m^2$. Similar values were obtained when the *Erica* seedlings were modeled until 2012, suggesting that, although it had less regeneration capacity than *Calluna*, *Erica* was still able to recover after fire disturbance.

In contrast, when all the studies years (2007–2012) are considered, the *Calluna* seedling models show a clear increasing trend during 2007 and a slight decrease in 2008 until reaching around 70 seedlings per $m^2$, with a decrease in the seedling number in 2012. Thus, this trend was expected considering how the *Calluna* revegetation is produced and especially considering two premises: (1) There was a pulse of germination in 2007 that covered most of the space and, afterwards, the seedlings were slowly outcompeted by others, reducing the number of seedlings, (2) then, the values were somehow stabilized around 70 seedlings, with a slight decrease until 30 seedlings after 5 years. In any case, the seedling values per $m^2$ are adequate for recovery of the previous vegetation biomass.

Interestingly, the lack of clear differences between fertilized and not fertilized treatments in the seedling establishment of both species suggests that the fertilization effect was not significant based on the final seedling numbers. Subsequent trials in the same areas [18] have also shown that *Calluna* heathlands at the low-latitude distribution limit are resistant to cumulative (9-year) high N (56 kg N ha$^{-1}$ year$^{-1}$ plus background deposition) inputs, at both the building/young and mature growth stages, regarding plant functional group relative abundances, species composition, and richness. Meyer-Grünefeldt et al. [37] also suggested that analyses of plant growth responses to global change should include multifactorial approaches with a focus on different populations across a species distribution range.

## 5. Conclusions

The results of this study provide considerable insight into the role of fire regarding whether a post-burn soil seed bank can restore *Calluna*-dominated vegetation or not, which is related to the restoration potential of the soil seed bank after wildfires. Our results suggest that, although there were slight vegetation compositional differences at the beginning, in the short term after burning, the heathland community recovered its main characteristic species such as *Calluna* and *Erica*, which is fundamental to maintain the ecosystem structure. The other species, mostly herbs, are expected to recover in the environment in the years to come, especially *Nardus, Deschampsia,* and *Agrostis* present in the seed bank, and therefore, renew the plant community [38].

The soil seed bank plays a fundamental role as a seed reservoir for the recovery of heathland ecosystems in the Cantabrian Mountain range. It is true that, after fire, the appearance of seedlings of the dominant species is delayed in time, possibly due to the short

vegetative period in these high mountain areas that reduces the windows of opportunity to germinate [39]. However, the after-fire seedling germinations are produced in sufficient numbers for the heath to return to its original state. Based on our results, it seems that the traditional management by prescribed fires of the heaths in the Cantabrian Mountain range rejuvenates these ecosystems, guaranteeing their survival in the long term [12]. However, we need to consider that these heathland ecosystems depend for their maintenance on their reservoirs in the soil seed bank, since it is their main protection strategy after fire events.

**Author Contributions:** Conceptualization, L.V. and J.G.A.; methodology, L.V. and J.G.A.; software J.G.A.; validation, L.V. and J.G.A.; formal analysis, J.G.A.; investigation, L.V., L.C., J.L.F.R. and J.G.A.; resources, L.V. and L.C.; data curation, L.V. and J.G.A.; writing—original draft preparation, J.G.A.; writing—review and editing, L.V., L.C., J.L.F.R. and J.G.A.; visualization J.G.A.; supervision, L.V.; project administration, L.V. and L.C.; funding acquisition, L.V. and L.C. All authors have read and agreed to the published version of the manuscript.

**Funding:** This work was supported by the Regional Government of Castilla and León (project, LE039A05); the Spanish Ministry of Education and Science (project, CGL2006-10998-C02-01/BOS.) and the Regional Government of Castilla and León (project, LE021A08). J.G.A. was supported by a Ramon y Cajal fellowship (RYC-2016-20528).

**Data Availability Statement:** Data will be made available upon reasonable request to the corresponding author.

**Acknowledgments:** We thank all people that have worked in this experiment.

**Conflicts of Interest:** The authors declare no conflict of interest.

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
