# Peer review of "The Soil Seed Bank Role in Mountainous Heathland Ecosystems after Fire and Inorganic Nitrogen Fertilization"

_forests, doi:10.3390/f14020226_

Round 1
Reviewer 1 Report
Dear Editor and Authors,
I have read the manuscript "The Soil seed bank role in mountainous heathland ecosystems after fire and inorganic nitrogen fertilization".
The authors examined regeneration dynamics associated with the disturbances to which Calluna vulgaris-dominated heathlands are currently exposed (i.e., N deposition, fire exclusion, and decline in sheep grazing) in the Cantabrian Mountains. They characterized plant species composition and soil seed bank in three heathlands in Spain.
Their results suggest that shortly after burning, the heathland community recovers its main characteristic species, such as Calluna and Erica, which are essential for maintaining ecosystem structure The note is well written. The manuscript is very interesting.
I suggest adding information about the "history" of grazing in the study area. Perhaps the authors know the history of use of this site. If grazing has occurred there, it makes sense to mention it in the materials and methods and in the discussion. Because grazing has a positive effect on building and enriching the soil seed bank, which is an important fact.
What I am missing is information on how long authors observed seedling emergence. One season? Or more? (lines 171-177).
Author Response
Dear Editor and Authors,
Point 1: I have read the manuscript "The Soil seed bank role in mountainous heathland ecosystems after fire and inorganic nitrogen fertilization".
The authors examined regeneration dynamics associated with the disturbances to which Calluna vulgaris-dominated heathlands are currently exposed (i.e., N deposition, fire exclusion, and decline in sheep grazing) in the Cantabrian Mountains. They characterized plant species composition and soil seed bank in three heathlands in Spain.
Their results suggest that shortly after burning, the heathland community recovers its main characteristic species, such as Calluna and Erica, which are essential for maintaining ecosystem structure The note is well written. The manuscript is very interesting.
Response 1: We are glad about these comments.
Point 2: I suggest adding information about the "history" of grazing in the study area. Perhaps the authors know the history of use of this site. If grazing has occurred there, it makes sense to mention it in the materials and methods and in the discussion. Because grazing has a positive effect on building and enriching the soil seed bank, which is an important fact.
Response 2: Ok, following the reviewer comments we have described little bit more the grazing history of the study area. Now it reads:
L46-59: “Traditionally these lands were used during the summer for grazing by transhumant flocks of sheep from the south of Spain and by flocks from neighboring villages. In contrast, traditional burnings were carried out in those areas where the heaths were very dense preventing the entry of sheep and cattle [5]. Because of these traditional management, Calluna-heathlands poses great floristic richness adapted to recurrent disturbances, being a well-known biodiversity hotspot [6]. For instance, in the heathlands of the Cantabrian range we can find more than 80 plant species, with Calluna vulgaris and Erica tetralix being the dominant species [7]. However, the abandonment of rural areas in the last decades is threatening the Spanish Calluna-heathlands by three forces: firstly, common to other European regions there has been an intense decrease in the number of cattle feeding in mountain areas [8], for instance in 1832 there were 551,032 transhumant sheep in the Province of León while in 2000 there were only 10,166 sheep [9]. Secondly, the type of livestock grazing has changed; nowadays there has been an increase in the number of cows and horses and a reduction of sheep [9].”
Point 3: What I am missing is information on how long authors observed seedling emergence. One season? Or more? (lines 171-177).
Response 3: Ok, we have detailed the duration of the seedling emergence experiments and even the soil seed bank emergence test. Now it reads:
L158-162: “At the same time, after the burning treatment application (burned and burned plus fertilized plots) we developed an emergence test counting the number of C. vulgaris and E. tetralix seedlings during the vegetative period (June – October) from 2005 to 2007. However, in 2008, 2009 and 2012 the emergence sampling was reduced to June and September.”
L186-189: “The soil seed bank emergence experiments last one year each (2004 and 2008).”

Reviewer 2 Report
Why did it take such a long time to publish the results (over 8 years)?
I think it would be interesting to present the method of counting seeds in the soil, as well as the method of determining their germination.
I think it would be useful to complete the following:
On line 118: under "bulk inorganic N deposition from 2011 to 2014" it should also be mentioned over which period this multi-year average is calculated.
l. 141: the year when these data were determined should also be mentioned in the title of the table. idem at l. 233 and 301.
Author Response
Dear Editor and Authors,
Point 1: Why did it take such a long time to publish the results (over 8 years)?
Response 1: Its good question, in this area we develop a lot of ecosystem experiments since 2004 until nowadays. Thus, we have a lot of datasets that are been transformed in SCI paper with time. In this case, although the experiment started in 2004 until 2013, we did not get all the data needed. Therefore, we have been publishing related papers until we get time to handle this dataset and write it. We are very happy with the results and the conclusions shown.
Point 2: I think it would be interesting to present the method of counting seeds in the soil, as well as the method of determining their germination.
Response 2: Ok, we have described in detail how it was the seeds counting process. Now it reads:
L154-165: “At the same time, after the burning treatment application (burned and burned plus fertilized plots) we developed an emergence test counting the number of C. vulgaris and E. tetralix seedlings during the vegetative period (June – October) from 2005 to 2007. For that we used the permanent sampling plots of each block marking and counting the seedlings of the two species that germinate and survive. Nevertheless, in 2008, 2009 and 2012 the emergence sampling was reduced to June and September.”
L183-189: “The soil seed bank emergence experiments last one year each (2004 and 2008). Emerging seedlings were identified, counted and removed, when the identification was un-certain the seedling was transplanted into a pot and grow on until an accurate identification could be made.”
I think it would be useful to complete the following:
Point 3: On line 118: under "bulk inorganic N deposition from 2011 to 2014" it should also be mentioned over which period this multi-year average is calculated.
Response 3: Ok, but the data reported here is from models that simulate the nitrogen deposition. In our case we are reporting the deposition in 5 years average. In any case for simplicity, we have included a brief comment in the text.
L129-131: “For instance, bulk inorganic N deposition from 2011 to 2014 was 4.6 kg N ha−1 yr−1, and total N deposition ranges between 7.5 and 15 kg N ha−1 yr−1 (in 5 years averages), according to the EMEP and CHIMERE models for Spain [23]”
Point 4: l. 141: the year when these data were determined should also be mentioned in the title of the table. idem at l. 233 and 301.
Response 4: Ok, we have clarified the data when all the measurements were collected or calculated.
